# The Symptomatic Calcification and Ossification of the Ligamentum Flavum in the Spine: Our Experience and Review of the Literature

**DOI:** 10.3390/jcm13010105

**Published:** 2023-12-24

**Authors:** Misao Nishikawa, Masaki Yoshimura, Kentaro Naito, Toru Yamagata, Hiroyuki Goto, Mitsuhiro Hara, Hiromichi Ikuno, Takeo Goto

**Affiliations:** 1Department of Neurosurgery, Moriguchi-Ikuno Memorial Hospital, 6-17-33 Satanakamachi, Moriguchi City 570-0002, Osaka, Japan; punchdrunkard830@yahoo.co.jp (T.Y.); mhara-senri@kra.biglobe.ne.jp (M.H.); 2Department of Neurosurgery, Osaka Metropolitan University Graduate School of Medicine, 1-4-3 Asahimachi, Abeno-ku, Osaka City 545-8595, Osaka, Japan; 7110ken622@omu.ac.jp (K.N.); gotot@omu.ac.jp (T.G.); 3Department of Neuropathology, Yao Tokusyukai General Hospital, 1-17, Wakakusacho, Yao City 581-0011, Osaka, Japan; masaking33@gmail.com; 4Department of Neurosurgery, Osaka Saiseikai Nakatsu Hospital, 2-10-39, Kita-ku, Osaka City 530-0012, Osaka, Japan; gotosanpoints@gmail.com; 5Department of Radiology, Moriguchi-Ikuno Memorial Hospital, 6-17-33 Satanakamachi, Moriguchi City 570-0002, Osaka, Japan; hi@koudoukai.jp

**Keywords:** calcification of the ligamentum flavum, ossification of the ligamentum flavum, pathogenesis, clinical feature, radiological finding, surgical management

## Abstract

Introduction: We report our experience regarding the clinical features and pathological findings of the calcification of the ligamentum flavum (CLF) and ossification of the ligamentum flavum (OLF) in the spine. In addition, we reviewed the previous studies on CLF and OLF to enhance the understanding of these conditions. Materials and Methods: We compared the clinical, radiological, and histopathological features of CLF and OLF. Results: In CLF, a computed tomography (CT) scan showed egg-shaped or speck-like calcification in the ligamentum flavum. Magnetic resonance (MR) imaging demonstrated spinal cord compression due to a thickened ligamentum flavum, which appeared as a low-intensity mass. Pathological findings demonstrated fused islands of calcification resembling sand-like calcification. In OLF, CT showed beak-like ossification extending into the intervertebral foramen. MR imaging demonstrated spinal cord compression by a low-intensity mass. Pathological findings revealed laminar ossification of LF with chondrocytes near the calcification and laminar hyaline cartilage. Conclusions: CLF and OLF appear to be distinct entities based on their clinical, neuroradiological, histopathological, and pathogenetic features. We suggest that the causes of CLF include both metabolic and dystrophic factors, while the pathogenesis of OLF is characterized by enchondral ossification induced by a genetic cascade triggered by shearing/tension stress.

## 1. Introduction

Calcification of the ligamentum flavum (CLF) is a disease that spinal surgeons often encounter. Some studies have categorized CLF as distinct from the ossification of the ligamentum flavum (OLF) [1,2,3,4,5,6,7,8,9,10,11]. The clinical, radiological, and histopathological features and surgical management of CLF are well-known [1,2,4,7,11,12,13,14,15,16]. However, its pathogenesis is relatively less clear [4,5,7,8,12,13,14,15].

OLF is frequently encountered by spinal surgeons, and numerous previous studies have described its clinical and radiological features and surgical treatment [17,18,19,20,21,22,23,24,25,26,27,28,29,30,31,32,33,34,35,36,37,38]. Few studies in the past have also described the histopathological features of OLF [18,20,21,22,23,24,30], but recent experimental studies have explored its pathogenesis using molecular and biological aspects [20,21,22,23,24,39,40,41,42,43,44]. 

We retrospectively encountered 15 cases of CLF and 20 cases of OLF that required surgical treatment and histopathological analysis of excised specimens. In this study, we retrospectively reviewed and validated our cases as well as the previous literature to compare CLF and OLF in terms of their clinical, radiological, and histopathological features and pathogenesis. In this study, we compared the distinct characteristics of CLF and OLF to enhance our understanding of these conditions, particularly in the context of therapeutic considerations, such as surgical management.

## 2. Materials and Methods

### 2.1. Materials

A total of 15 cases of CLF and 20 cases of OLF with myelopathy were treated at Moriguchi-Ikuno Memorial Hospital and Osaka Metropolitan University Hospital between April 2010 and March 2022. All cases were followed for more than 1 year after surgery.

### 2.2. Methods

#### 2.2.1. Clinical Features: Ex, Age, Prevalence, Location, Associated Diseases, Neurological Symptoms and Signs, and Neuroradiological Examination (Table 1, Table 2, Table 3 and Table 4)

Sex, age, prevalence, location, and associated diseases were evaluated in each CLF and OLF. Pre- and post-operative neurological symptoms and signs were assessed. Neurological signs and activities of daily living (ADL) were evaluated using the Japanese Orthopaedic Association (JOA) score in cervical spondylotic myelopathy [45]. In thoracic and thoraco-lumbar spine lesions, the points for upper extremities were excluded, while in lumbar spine lesions, the points for upper extremities and trunk were excluded. This resulted in a maximum score of 17 points for cervical and cervico-thoracic spine lesions, 11 points for thoracic and thoraco-lumbar spine lesions, and 9 points for lumbar spine lesions. The recovery rate (RR%) was calculated as follows: <(postoperative score − preoperative score)/(maximum score − preoperative score)> × 100(%). Pre-operative whole spinal dynamic X-rays, two-dimensional (2D) computed tomography (CT), and magnetic resonance (MR) imaging were performed. Post-operative neurological symptoms, signs, and ADL were checked up, and spinal dynamic X-rays, 2D-CT, and MR imaging of lesions were performed every 6 months. There was no change in the treatment protocol or diagnostic criteria between April 2010 and March 2022.

**Table 1 jcm-13-00105-t001:** Clinical features of CLF and OLF: sex, prevalence, location, and associated diseases.

	CLF	OLF
**Sex: Male/Female**	4/11	15/5
**Age: years old (mean ± 1SD)**	71~91 (79.7 ± 5.2)	34.4~82.7 (60.1 ± 15.2)
30~39	0	2
40~49	0	5
50~59	0	2
60~69	0	2
70~79	8	7
80~89	6	2
90~99	1	0
**Prevalence/spondylotic change**		
Followed by surgery (%)	15/1800 (0.8)	20/1800 (1.1)
**Location**		
Cervical spine	12	0
Cervico-thoracic spine	3	5
Thoracic spine	0	12
Upper	0	4
Lower	0	8
Thoraco-lumbar spine	0	3
Lumbar spine	0	0
**Associatated disease**		
HTN	15	18
Dyslipidemia	14	17
DM	12	15
Obesity	12	14
Hypothyroidism	7	3
COPD	4	2
Uric acidemia	2	2
CKD	2	1
CHF	2	1
Rheumatoid arthritis	2	1

**Abbreviations:** CLF = calcification of ligamentum flavum, OLF= ossification of ligamentum flavum, SD = standerd deviation, HTN = hyoertenstion, DM = diabetus melitus, COPD = chronic obstructive pulmonary disease, CKD = chronic kidny disease, CHF = chronic heart failure.

**Table 2 jcm-13-00105-t002:** Neurological symptoms, signs, and JOA score of CLF and OLF.

	CLF	OLF
**Neurological symptoms**		
Gradually progresive gait disturbance	15	20
Motor weakness of the expremities	15	20
Impairment of fine movement of fingers	15	5
Focal numbness and/or dysthesia at the extremities	15	15
Radicular pain at the lesion levels	0	7
**Neurological signs**		
Transverse myelopathy at the cord level	10	6
Tetraparesis	7	3
Paraparesis	3	3
Sensory disturbance	10	6
Brown-Sequard type myelopathy at the cord level	5	3
Conus medullaris syndrome	0	7
Roots signs at lumbar levels	0	4
**JOA score **(average ± SD) **(Pre-Op.)**		
Cervical, cervico-thoracic spine (full: 17 points)	5~10 (7.6 ± 2.1)	4~7 (3.3 ± 2.2)
Thoracic, thoraco-lumbar spine (full: 11 points)	N.A.	3~5 (3.8 ± 1.3)
Lumbar spine (full: 9 points)	N.A.	3~5 (3.3 ± 1.0)

**Abbreviations:** CLF = calcification of ligamentum flavum, OLF= ossification of ligamentum flavum, JOA = Japanese Orthopardic Association, N.A. = not aplication.

**Table 3 jcm-13-00105-t003:** Neuroradiological findings of CLF and OLF.

	CLF	OLF
**Radiological associated condition**		
Spndylotic change	15	17
DNSC	5	7
Disc hernia	4	7
OPLL	3	4
DISH	0	4
ASH	0	3
**2D-CT scan images**	egg shape high density	semilunar & laminar high density
	salt and pepper like high density	involed inter vertebaral foramen
	diffuse and speck like high density	large round high density
**MR images**	various size	semilunar & laminar low intensity
	round low intensity mass	involed inter vertebaral foramen
		large round low intensity
		and its attached to the dura matter
		high intensity spots in low intensity

**Abbreviations:** CLF = calcification of ligamentum flavum, OLF = ossification of ligamentum flavum, DNSC = developmental narrow spinal canal, OPLL= Ossification of posterior lomgituidinal ligament. DISH= diffuse idiopathic skeletal hyperostosis, ASH = ankylotic spinal hyperostosis.

**Table 4 jcm-13-00105-t004:** Surgeries and outcomes of CLF and OLF.

	CLF	OLF
**Surgery**		
Laminectomy	2	12
Laminoplasty	8	5
Laminectomy with PLF	5	3
**Outcome**		
**Improvement of neurological symptoms (improvement rate (%))**		
Gradually progresive gait disturbance	15/15 (100)	17/20 (85)
Motor weakness of the expremities	15/15 (100)	17/20 (85)
Impairment of fine movement of fingers	13/15 (87)	4/5 (80)
Focal numbness and/or dysthesia at the extremities	11/15 (73)	8/15 (53)
Radicular pain at the lesioin levels	0	6/7 (86)
**Improvement of neurological signs (improvement rate (%))**		
Transverse myelopathy at the cord level	10/10 (100)	15/15 (100)
Tetra-paresis	7/7 (100)	3/3 (100)
Para-paresis	3/3 (100)	3/3 (100)
Sensory disturbance	7/10(70)	4/6 (67)
Brown sequred type myelopathy at the cord level	11/15 (73)	5/6 (83)
Conus medullaris syndrome	0	7/11 (64)
Roots signs	0	8/10 (80)
**Post-Op. JOA score (** **average ± SD)**		
**Recovery rate R.R. (%) (average ± SD)**		
Cervical, cervico-thjoracic spine (full: 17 points)	14~17 (15.8 ± 1.5)	13~17 (15.4 ± 2.5)
R.R. (%) (average ± SD)	67.7–85.7 (72.0 ± 5.4)	66.7–87.5 (71.3 ± 11.5)
Thoracic, thoraco-lumbar spine (full: 11 points)	N.A.	8~11 (9.8 ± 2.6)
R.R. (%) (average ± SD)	N.A.	58.7–78.7 (68.3 ± 10.1)
Lumbar spine (full: 9 points)	N.A.	7~9 (8.8 ± 1.4)
R.R. (%) (average ± SD)	N.A.	58.3–79.5 (69.3 ± 10.7)

**Abbreviations:** CLF = calcification of ligamentum flavum, OLF = ossification of ligamentum flavum, PLF = posterior lateral fixation, JOA = Japanese Orthopadic association, SD= standerd deviation, HA = hydroxyapatite, N.A.= not aplication.

#### 2.2.2. Surgical Indication and Procedure (Figure 1 and Figure 2, Table 4)

Surgery was indicated for patients with neurological signs of progressive myelopathy and/or radiculopathy. Posterior decompression by laminectomy or laminoplasty was performed under somatosensory evoked potential monitoring [46,47]. 

**CLF:** Conventional laminectomy or en-bloc laminectomy and laminoplasty using basket-type titanium plates were performed after the dissection of the CLF (Figure 1A,B) [46,47,48]. For cases with instability, single-stage posterior decompression (laminectomy) and posterior lateral fixation (laminectomy with posterior lateral fixation [PLF]) using instrumentation were performed (Figure 1C,D) [49]. In most of the cases of CLF, there was no adhesion between the ligamentum flavum and the dura mater, and calcification did not extend to the intervertebral foramen. Therefore, en-block laminoplasty was generally considered the primary choice. For instances where calcification was extensive, such as when calcification extended to the intervertebral foramen, laminectomy and posterior lateral fixation were performed. 

**Figure 1 jcm-13-00105-f001:**
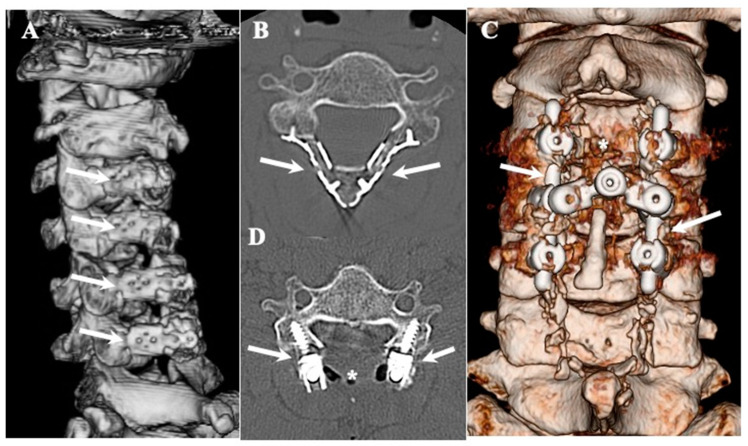
**Three-dimensional and two-dimensional CT images of CLF.** Three-dimensionally reconstructed CT image showing changes after en-bloc expansive laminectomy and laminoplasty using basket-type titanium plates (arrows) (**A**) and axial image (**B**). Three-dimensionally reconstructed CT image showing changes after single-staged posterior decompression (asterisks) and posterior lateral fixation using lateral mass screws and rods (arrows) (**C**) and axial image (**D**).

**OLF:** In the cases of OLF, instances where the ligamentum flavum adhered to the dura mater were not uncommon, and ossification extending to the intervertebral foramen was also frequently observed. Therefore, we adopted laminectomy following the approach outlined in Figure 2. In cases where ossification did not extend to the intervertebral foramen nor adhered to the dura mater, we attempted en-block laminoplasty, considering the need to reconstruct posterior supporting tissue for long-term outcomes. A gutter with a similar width to the spinal canal was created, and the OLF was drilled out from the laminae. In cases where OLF was severely adherent to the dura mater, part of the OLF was intentionally left behind (Figure 2) [50,51]. For cases with instability, single-stage laminectomy with PLF was performed [49]. 

**Figure 2 jcm-13-00105-f002:**
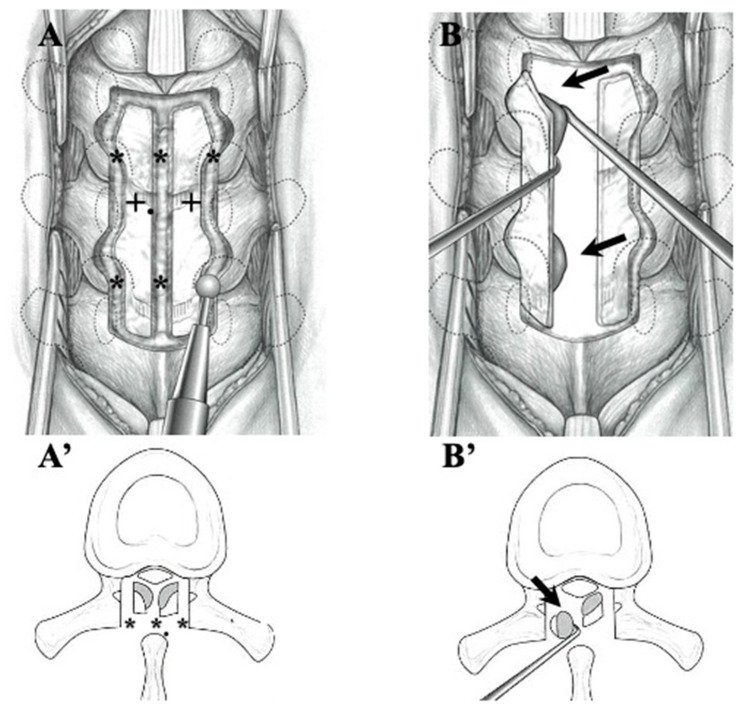
**Illustrations demonstrating laminectomy for OLF in the thoracic spine.** After thinning the lamina (black cross), cranial, caudal, lateral, and midline garters were made by drilling (black asterisks) (**A**,**A’**). The OLF was carefully dissected from the dura matter and removed (black arrows) (**B**,**B’**).

#### 2.2.3. Histopathological Examination (Table 5)

Specimens obtained during surgery were evaluated using hematoxylin–eosin (HE) staining, Alcian blue Periodic acid–Schiff (PAS) staining under an optical microscope, and hydroxyapatite (HA) and calcium pyrophosphate dihydrate (CPPD) staining under an electron microscope.

**Table 5 jcm-13-00105-t005:** Histopathological findings of CLF and OLF.

CLF	OLF
HA and/or CPPD deposition	laminar ossification
Island of calcification	hyaline cartilage
calcification not closing the dura mater	adjacent chondrocytes
the surface of the dura mater: intact	osteoblasts and bone marrow
	chondrocytes and osteoblasts closing ossification of dura mater
	ossification invades into dura mater

**Abbreviations:** CLF = calcification of ligamentum flavum, OLF= ossification of ligamentum flavum, HA= hydroxyapatite, CPPD= calcium pyrophosphate dihydrate,.

## 3. Results

### 3.1. Clinical Features: Sex, Prevalence, Location, and Associated Diseases (Table 1)

**CLF:** Among the 15 CLF cases, there were four males and 11 females, indicating a predominance of females. The age at the time of operation was 71–91 years (average age: 79.7 ± 5.2 years). The lesions were mainly located in the middle or lower cervical spine or upper thoracic spine. Of our 1800 patients who presented to MIMH and OMUH between April 2010 and March 2022 for spinal surgery due to neurological symptoms and spondylotic changes, 15 (0.8%) had CLF. All patients had hypertension (HTN), 14 had dyslipidemia, 12 had diabetes mellitus (DM), 12 had severe obesity, 7 had hypothyroidism, 4 had chronic obstructive pulmonary disease (COPD), 2 had uric acidemia, 2 had chronic kidney disease (CKD), 2 had chronic heart failure (CHF), and 2 had rheumatoid arthritis (RA) (Table 1).

**OLF:** Among the 20 OLF cases, 15 were males, and 5 were females, indicating a male predominance. The age at the time of the operation was 34.4–82.7 years (average age: 60.1 ± 15.2 years). Although there was a wide age range, the fourth and seventh decades were the most common. The lesions were located lower cervical spine to the upper thoracic spine in five cases, the upper thoracic spine in four cases, the lower thoracic spine in eight cases, and the thoraco-lumbar to the lumbar spine in three cases. Of our 1800 patients who presented to MIMH and OMUH between April 2010 and March 2022 for spinal surgery due to neurological symptoms and spondylotic changes, 22 (1.2%) had OLF. The comorbidities included HTN in 18 cases, dyslipidemia in 17 cases, DM in 15 cases, 14 had severe obesity, hypothyroidism in 3 cases, COPD in 2 cases, uric acidemia in 2 cases, CKD in 1 case, CHF in 1 case, and (RA) in 1 case (Table 1).

### 3.2. Neurological Symptoms, Signs, and JOA Score (Table 2)

**CLF:** The neurological symptoms included gradually progressive gait disturbance in 15 cases, motor weakness of the extremities in 15 cases, impairment of fine motor movements of fingers in 15 cases, sensory disturbances in 15 cases, and focal numbness and/or dysesthesia at the extremities in 11 cases. All cases developed myelopathy at the lesion site. Tetra paresis developed in 7 cases, paraparesis developed in 3 cases, and sensory disturbance developed in 10 cases. The neurological signs of Brown-Sequard-type myelopathy developed in 5 cases. The pre-operative JOA score was 5–10 (average: 7.6 ± 2.1) (Table 2).

**OLF:** The neurological symptoms included gait disturbances in 20 cases, motor weakness in the extremities in 20 cases, and impairment of fine motor movements of fingers in 5 cases with OLF at the cervico-thoracic lesion. Focal numbness and/or dysphasia at the extremities in 7 cases caused radicular pain at the lesion level. In terms of neurological signs, all cases developed myelopathy at the lesion site, tetraparesis developed in three cases, paraparesis developed in three cases, and sensory disturbance developed in six cases. In total, 3 cases had Brown-Sequard type myelopathy, 11 cases had conus medullaris syndrome, and 10 cases had signs of lumbar root damage. The pre-operative JOA score in cervical and cervico-thoracic spine OLF was 4–7 (average: 3.3 ± 2.2). JOA score in thoracic and thoraco-lumbar spine OLF was 3–5 (average: 3.8 ± 1.3). JOA score in lumbar spine OLF was 3–5 (average: 3.3 ± 1.0) (Table 2).

### 3.3. Neuroradiological Findings (Figure 3, Figure 4, Figure 5 and Figure 6; Table 3)

**CLF:** All cases exhibited spondylotic changes. Developmental narrowing (anterior-posterior diameter of spinal canal < 12 mm) of the spinal canal was observed in five cases, disc herniation in four cases, and ossification of the longitudinal ligament in three cases. Two-dimensional (2D)-reconstructed CT images demonstrated “multiple islands of speck-like and egg-shaped” high-density mass, “diffuse and speck-like” high-density masses, and/or “egg-shaped” high-density masses in the ligamentum flavum (Figure 3). In three cases, there was ossification of the longitudinal ligament (Table 3). MR T2-weighted images showed these mass lesions as areas of low intensity or isointensity, with compression of the spinal cord. The presence of both low-intensity and isointense regions suggested a combination of calcification and ligamentum tissue (Figure 4). In five cases, dynamic X-rays of the cervical spine demonstrated instability of joints (Table 3).

**Figure 3 jcm-13-00105-f003:**
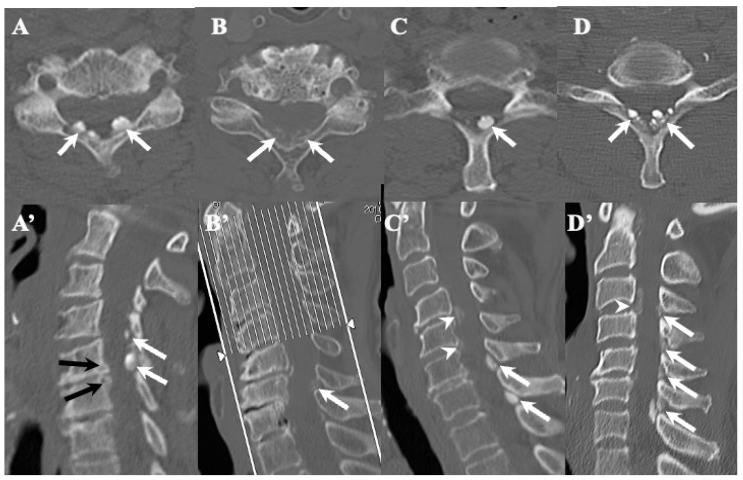
**Two-dimensional CT images of CLF.** Two-dimensionally reconstructed CT scan axial image (**A**) and sagittal image (**A’**) CT scan demonstrating “egg shape” high-density mass contacting with laminae (arrows) and posterior osteo-spur at the C5/6 intervertebral level (black arrows). Two-dimensionally reconstructed CT scan axial image (**B**) and sagittal image (**B’**) demonstrating “diffuse and speck like” high-density mass contacting with laminae in ligamentum flavum (white arrows). Two-dimensionally reconstructed CT scan axial image (**C**) and sagittal image (**C’**) demonstrating “egg shape” high-density mass in ligamentum flavum on the left side in (**C**) and along the ligamentum flavum in (**C’**) (white arrows), ossification of posterior longitudinal ligament at the C4/5, 5/6 (white arrowheads), and listhesis at C3/4 intervertebral level. Two-dimensionally reconstructed CT scan axial image (**D**) and sagittal image (**D’**) demonstrating “multiple islands of speck like and egg shape” high-density mass in ligamentum flavum (arrows) and ossification of longitudinal ligament at C3/4 (white arrowhead).

**Figure 4 jcm-13-00105-f004:**
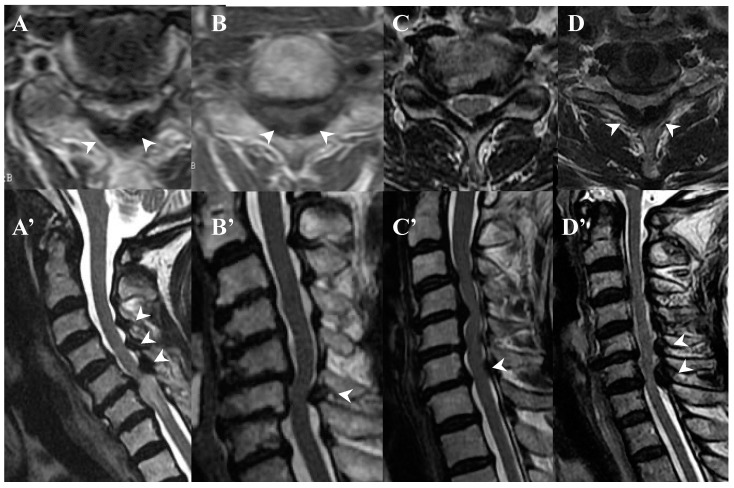
**MR images of CLF.** MR T2-weighted axial images (**A**–**D**) and sagittal images (**A’**–**D’**) from the same patient showing low intensity (arrow heads).

**OLF:** There were spondylotic changes in 17 cases, developmental narrowing of the spinal canal in 7 cases, disc herniation in 7 cases, ossification of the longitudinal ligament in 4 cases, diffuse idiopathic skeletal hyperostosis (DISH) in 4 cases, and ankylotic spinal hyperostosis (ASH) in 3 cases [31,51,52]. Two-dimensionally reconstructed CT images showed beak-like ossification extending into the intervertebral foramen (Figure 5) (Table 3). MR T1-weighted and T2-weighted images demonstrated low-intensity masses in the dorsal spinal canal, which compressed the spinal cord. In some cases, there were high-intensity spots, suggesting the formation of bone marrow (Figure 6). In three cases, dynamic X-rays of thoraco-lumbar spine demonstrated joint instability (Table 3).

**Figure 5 jcm-13-00105-f005:**
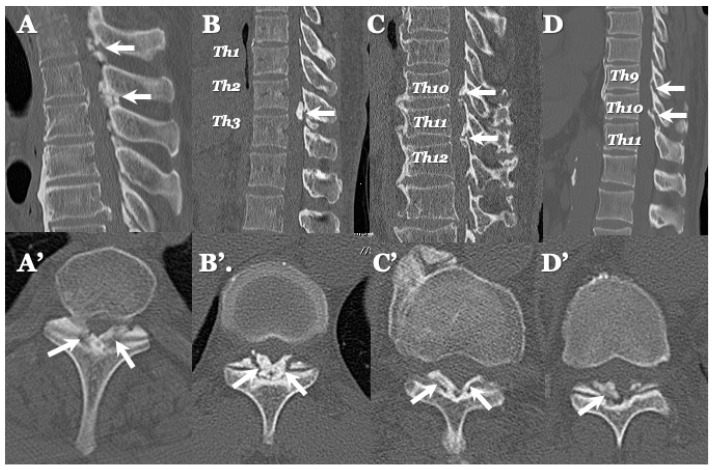
Two-dimensional **CT images of OLF.** Two-dimensionally reconstructed CT scan sagittal images (**A**–**D**) image and axial image (**A’**–**D’**) showing beak-like ossification extended to intervertebral foramen (white arrows).

**Figure 6 jcm-13-00105-f006:**
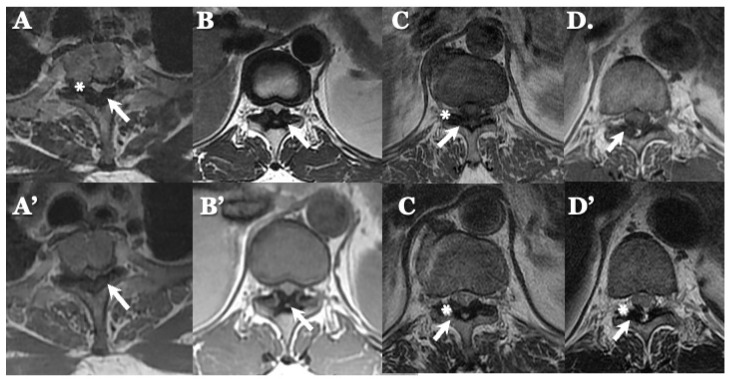
**MR images of OLF.** MR T1-weighted axial images (**A**–**D**) and T2-weighted axial images (**A’**–**D’**) demonstrating low-intensity mass in dorsal spinal canal compressing spinal cord (arrows) and high-intensity spots (white asterisks).

### 3.4. Surgeries and Outcomes (Table 4)

**CLF:** Laminectomy was performed in two cases, laminoplasty was performed in eight cases, and single-stage laminectomy with PLF was performed in five cases with instability (Figure 1 and Figure 2) [46,47,48,49]. Of the eight cases that underwent expansive laminoplasty, five underwent en-block expansive laminoplasty using basket-type titanium miniplates with HA and collagen sponge (Figure 1) [48].

The gait disturbance improved in 15 cases (100%), motor weakness of the extremities improved in 15 cases (100%), impairment of fine motor movements of fingers improved in 13 cases, and focal numbness and/or dysesthesia at the extremities improved in 11 cases (73%). Tetraparesis improved in 7 cases (100%), paraparesis developed in 3 cases (100%), and sensory disturbance improved in 10 cases (70%). Brown-Sequard-type myelopathy improved in 11 cases (73%).

At 1 year post-operation, the JOA score was 14–17 (average: 15.8 ± 1.5), with RR% of 67.7–85.7% (average: 72.0 ± 5.4%). All cases were able to walk independently and perform daily activities without assistance (Table 4). There was no mortality or morbidity.

**OLF:** Laminectomy was performed in 12 cases, expansive laminoplasty was performed in 5 cases, and single-stage laminectomy with PLF was performed in 3 cases with instability [46,47,48,49]. Of the five cases that underwent expansive laminoplasty, three underwent en-block expansive laminoplasty using basket-type titanium miniplates with HA and collagen sponge (Figure 1) [49]. In terms of the intraoperative findings, OLF was in contact with the dura matter in 10 cases and adhered to the dura matter in 5 cases. The dura matter was opened by OLF resection and sutured using a 6–0 Nylone thread. The DuraGen^®^ was covered with fibrin glue.

The gait disturbance improved in 17 cases (85%), motor weakness of the extremities improved in 17 cases (85%), impairment of fine motor movements of fingers improved in 4 cases (80%) with OLF at cervico-thoracic lesion, focal numbness and/or dysthesia at the extremities improved in 8 cases (53%), and the radicular pain at the lesion level improved in 6 cases (86%). Tetra paresis improved in 15 cases (100%), paraparesis improved in 3 cases (100%), sensory disturbance improved in 4 cases (67%), Brown-Sequard-type myelopathy improved in 5 cases (83%), conus medullaris syndrome improved in 7 cases (64%), and signs of lumbar root damage improved in 8 cases (80%).

At 1 year post-operatively, in cervical and cervico-thoracic spine lesions, the JOA score was 13–17 (average: 15.4 ± 2.5), with RR% of 66.7–87.5% (average: 71.3 ± 11.5). In thoracic, thoraco-lumbar spine lesions, the JOA score was 8–11 (average: 9.8 ± 2.6), with an RR% of 63.2–88.5% (average: 68.3 ± 10.1). In lumbar spine lesions, the JOA score was 7–9 (average: 8.8 ± 1.4), with an RR% of 63.4.–87.2% (average: 69.3 ± 10.7) (Table 4). Of the 20 cases, 17 were able to walk independently and perform daily activities without assistance, whereas 6 cases needed some assistance for daily activities at the time of discharge but showed improvement with rehabilitation and were able to perform daily activities independently after 6 months. There was no mortality or morbidity.

### 3.5. Histopathological Findings (Figure 7 and Figure 8; Table 5)

**CLF:** The ventral surface of the laminae with the ligamentum flavum showed a smooth surface of the ligamentum flavum intraoperatively. The ligamentum flavum was easily removed from the laminae. The ventral surface of the laminae with ligamentum flavum demonstrated islands of calcification and their fusion on being cut, resembling sand-like calcification. HE staining demonstrated these islands of calcification and their fusion but did not demonstrate chondrocytes or osteoblasts (Figure 7). In 12 cases, the presence of HA or CPPD was confirmed by electron microscopy.

**Figure 7 jcm-13-00105-f007:**
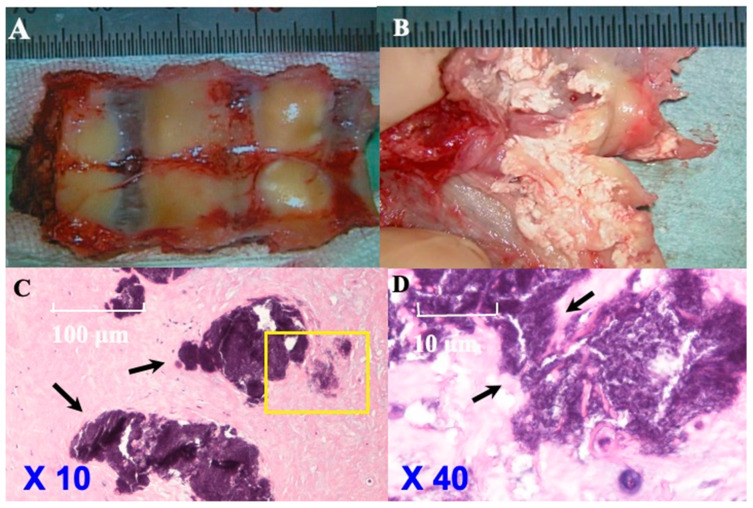
**Histopathological findings of CLF in illustrative case.** Ventral side of laminae with ligamentum flavum while in operation (**A**) showing the smooth surface of ligamentum flavum and easily removed ligamentum flavum from laminae. Ventral side of laminae with yellow ligament cut (**B**) demonstrating island of calcification and fusion of them and many sand-like calcifications. Hematoxylin–eosin (H&E) staining (original magnification: 10×, 40×) (Case 10) (**C**,**D** (the area of yellow box was enlarged)) showed islands of calcification and their fusion (black arrows). Chondrocytes and osteoblasts were not identified.

**Figure 8 jcm-13-00105-f008:**
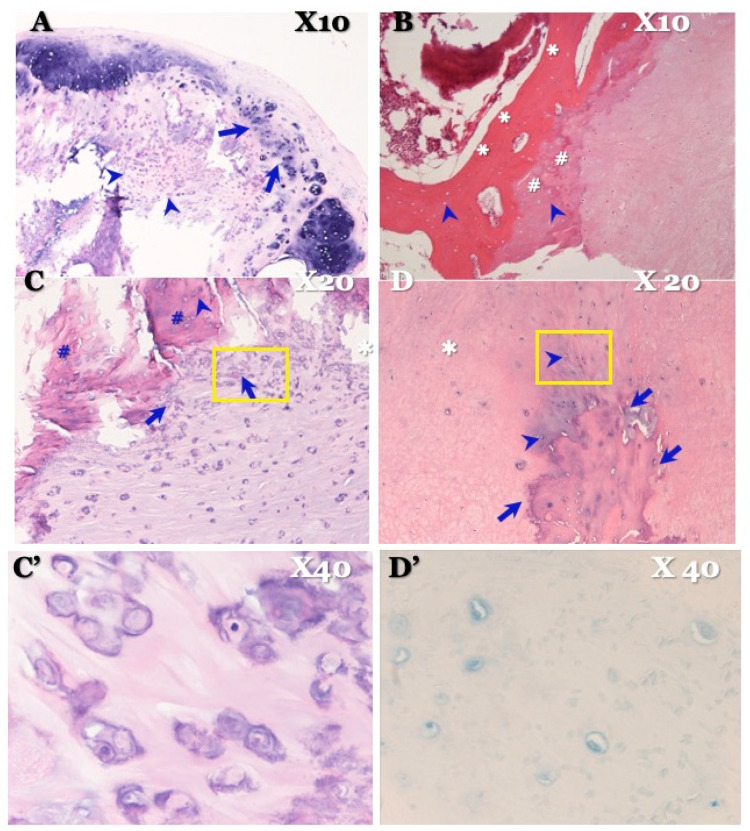
**Histopathological findings of OLF in illustrative case.** Hematoxylin and eosin (HE) stain × 4 picture (**A**) showing stratified calcium deposition standing in a line in yellow ligament attached to the dura mater (arrows) and group of chondrocytes (arrowheads). HE stain × 10 picture (**B**) showing stratified bony development (asterisks), hyaline cartilage (sharps), and osteoblasts in it (arrowheads). HE stain × 20 (**C**) showing chondrocytes near ossification (arrows), osteoblasts in ossification (arrowhead), and hyaline cartilage (sharps), which demonstrates chondral bone formation. HE stain high-magnification × 40 picture (**C’**) certifying chondrocytes. HE stain × 20 picture (**D**) showing ossification encroaches vertically to the dura matter (arrows). Chondrocytes were confirmed near ossification (arrowheads). Elastic fiber is intact (white asterisk), and chondral ossification grows, closing ossification of ligamentum flavum side. Alcian blue stain × 40 picture (**D’**) certifying chondrocytes in dura mater side.

**OLF:** HE staining showed stratified calcium deposition with a linear distribution in the ligamentum flavum, attached to the dura mater, along with groups of chondrocytes. HE staining also showed stratified bony development, hyaline cartilage, and osteoblasts, indicating chondral bone formation (Figure 8). Furthermore, HE staining exhibited ossification that encroached vertically into the dura matter. Chondrocytes were confirmed to be present near the ossification. The elastic fiber was intact, and chondral ossification was observed near the ossification of the ligamentum flavum. Alcian blue (PAS) staining demonstrated chondrocytes on the side of the dura mater (Figure 8). There were no inflammatory cells to suggest the presence of inflammation.

## 4. Discussion

### 4.1. Comparison of Clinical Features between CLF and OLF (Table 1 and Table 2)

**CLF:** All cases of CLF were elderly females aged ≥70 years, which is in line with previous studies, in which most cases were Asian females aged ≥70 years [1,2,3,4,5,6,7,8,9,10,11,12,13,14,15]. The lesion was predominantly located in the middle or lower cervical spine, consistent with previous reports [1,2,3,4,5,6,7,8,9,10,11,12,13,14,15,16]. In the present study, the prevalence of CLF requiring surgery in spondylotic changes was 15 (0.83%) out of the 1800 cases who presented to MIMH and OMUH between April 2010 and March 2022 for spinal surgery due to neurological symptoms and spondylotic changes. Notably, there are no previous reports of CLF [4]. The neurological symptoms and signs included gradually progressive myelopathy (Table 1 and Table 2) [1,2,3,4,5,6,7,8,9,10,11,12,13,14,15]. These symptoms and signs developed gradually due to the compression of the spinal cord by the calcified mass (Table 1 and Table 2). However, in some cases, neurological symptoms and signs developed rapidly due to disc herniation and instability.

**OLF:** Most of the male patients with OLF were in their fourth to sixth decades of life. The lesions were most commonly located in the cervico-thoracic and thoraco-lumbar regions [17,19,25,26,27,28,29]. The prevalence of OLF requiring surgery in spondylotic changes was 22 (1.2%) out of the 1,800 cases who presented to MIMH and OMUH between April 2010 and March 2022 for spinal surgery due to neurological symptoms and spondylotic changes, which was similar to the rates reported in previous studies [32,36,39]. The neurological symptoms and signs were characterized by gradually progressive myelopathy (Table 1 and Table 2) [32,36,37,39,51]. These symptoms and signs developed gradually due to ossified mass lesions compressing the spinal cord, but in some cases, rapid development occurred due to disc herniation and instability (Table 1 and Table 2). Most cases of CLF and OLF were associated with chronic diseases.

### 4.2. Comparison of Neuroradiological Findings between CLF and OLF (Figure 3, Figure 4, Figure 5 and Figure 6; Table 3)

A previous study compared CLF and OLF without considering MR imaging. Furthermore, changes in CT appearance over time were not evaluated, which prevented a detailed analysis of the differences between calcification and ossification, as well as their anatomical relationships [12]. In the present study, we compared the neuroradiological features of CLF and OLF.

**CLF:** A two-dimensionally reconstructed CT scan revealed “multiple islands of speck-like and egg-shaped” high-density mass and/or “diffuse and speck-like” high-density mass within the ligamentum flavum. MR T1- and T2-weighted images depicted these mass lesions as low intensity or isointense, with compression of the spinal cord. The presence of low-intensity and isointense regions suggested a combination of calcified regions and ligamentum tissue (Table 3, Figure 1 and Figure 2) [3,4,6,7,9,10,12,14,15,17]. 

**OLF:** Two-dimensionally reconstructed CT scan images showed beak-like ossification extending into the intervertebral foramen. MR T1-weighted and T2-weighted images demonstrated a low-intensity mass within the dorsal spinal canal compressing the spinal cord. In some cases, the presence of high-intensity spots suggested bone marrow formation (Figure 3 and Figure 4) [18,26,29,30,31]. In some cases, OLF was associated with DISH and ASH, indicating a more generalized ossification pattern (Table 3). Previous reports have reported the association of DISH and ASH in OLF, but the specific features of this association were unclear [51,52]. In both CLF and OLF, spondylotic changes were the most common associated condition, followed by disc herniation and instability. This suggested that chronic micro-movement and mechanical stress (traction, rotation, and compression) to the ligament were contributing factors (Table 3).

### 4.3. Surgical Treatments and Outcomes in CLF and OLF (Figure 1 and Figure 2; Table 4)

**CLF:** In most CLF cases, to prevent post-operative kyphosis in cases where calcification did not extend to the intervertebral foramen nor adhered to the dura mater, we attempted en-block laminoplasty. The ligamentum flavum, including CLF, can be removed without damage through en-block laminoplasty because the calcification is present inside the ligament and does not reach the surface of the ligament or cause adherence to the dura mater. However, caution is required when excising the ossified portion of the ligamentum flavum from the dura mater, as illustrated in Figure 2. In some instances, we deliberately left the calcification thin and chose not to detach it from the dura mater.

The high improvement rate was achieved by surgery. The motor weakness had completely recovered, with an improvement rate of 87% or 100%, and the sensory disturbance had an improvement rate between 70% and 75%. Particularly, the sensory disturbance of the dorsal column in the spinal cord had a low rate of improvement because the dorsal column had been directly compressed by the CLF. The post-operative JOA score RR was high due to the timely diagnosis and treatment (Table 4). These findings indicate that significant improvements can be achieved, particularly among the elderly [1,2,3,4,5,6,7,8,9,10,11,12,13,14,15,16,49]. 

**OLF:** In OLF, as an operative procedure, laminectomy was generally indicated. But we think that to prevent post-operative kyphosis, we should try to perform laminoplasty for the possible case where ossification does not extend to the intervertebral foramen nor adhere to the dura mater, as illustrated in Figure 5B,B’. Since most cases of OLF exhibit adhesion to and penetration into the dura mater, caution is required during OLF removal, as illustrated in Figure 2. In cases with adhesion to the dura mater, it may be necessary to carefully preserve the OLF or suture defects in the dura mater after the dissection of the OLF (Figure 2) [49,51]. 

The high improvement rate was achieved by surgery. The motor weakness had almost completely recovered by 80% or 100%, but the sensory disturbance had an improvement rate between 60% and 80%. Particularly, the neurological signs of the dorsal column in the spinal cord also had a low rate of improvement because the dorsal column had been directly compressed by the OLF. Significant improvements can be attained with timely diagnosis and treatment, as demonstrated in previous reports (Table 4) [53,54,55,56,57,58]. Patients with prolonged disease duration, other spinal diseases, or intramedullary high signals on sagittal T2-weighted images tend to have a worse prognosis [51,58]. Because OLF develops gradually, most patients do not develop neurological deterioration, leading to delayed diagnosis.

### 4.4. Pathogenesis of CLF and OLF Based on Histopathological Findings (Figure 7 and Figure 8; Table 5)

We compared the histopathological findings between CLF and OLF, including molecular biological characteristics. Conversely, the previous study did not examine the molecular biological characteristics [12]. In this study, we compared the pathological features of CLF and OLF. 

**CLF:** CLF has been found to not reach the surface of the ligamentum flavum in the current and past reports [12,13,14,15,16,17]. Histopathologically, calcification resembling an island, speck, or grains of sand was observed. From a chronological perspective, these calcifications grew larger over time, coalescing into larger lesions. Previous reports showed the presence of HA in the center of the calcified lesion, whereas CPPD crystals were found in the surrounding area. These crystals exhibited cube-, rhombus-, or rod-shaped structures, ranging in size from 3 to 30 microns, and were partially engulfed by neutrophils and macrophages [5,6,8,9,11,15,16,17]. These findings suggest that the pathogenesis of CLF involves inflammation and subsequent repair processes, primarily within ligamentous fibers. CPPD is involved in pseudogout and crystal-induced arthritis, suggesting that CLF is related to generalized calcinosis. Chondrocytes and osteoblasts were not observed in CLF, indicating that enchondral ossification does not occur. 

Calcification is categorized into metastatic or dystrophic [43]. Metastatic calcification occurs in normal tissue, whereas dystrophic calcification occurs in necrotic tissue after inflammation [43]. Heterotopic (metastatic and dystrophic) calcification, such as CLF and flavum ossification of the longitudinal ligament, is considered to involve the conversion of progenitor cells to osteogenic precursor cells as a result of cell-mediated interactions with the local tissue environment as well as oxygen tension, pH, micronutrient availability, and heterotopic ossification due to mechanical stimuli [43]. The phenomenon of calcification requires one or more of least three phenomena without endochondral cell: (a) local increase in the concentration of precipitable ionic species (e.g., Ca, PO_4_); (b) the presence of “nucleator,” that is, a promotor of crystal deposition; or (c) the modification or removal of macromolecules inhibiting crystal deposition [43]. Previous studies have suggested that the pathogenesis of CLF involves aging-related changes, endocrine disorders, and chronic stimulation [43]. Mechanical stimulation, particularly traction and rotation, causes ligament fiber injury and subsequent calcification during the inflammatory repair process, primarily within ligamentous fibers. Inside the calcification, CPPD concentrates, and HA is produced [7,8,11,12,13,14,15,16,17]. The lesion of calcification never connected to the epiphysis of the joints, and so the endochondral ossification never occurred. Because calcification progresses in the focal reaction and cell unit in ligamentum flavum, calcification occurs in the surrounding area of the injured collagen fiber, which is weak to tension and sharing stress and easy to be injured. 

**OLF:** The radiological and histopathological findings demonstrate that laminar ossification occurs along with the elastic fibers within the ligament. Microhistopathological findings showed adhesion and penetration into the dura matter, with ossification on the side of the dura mater [24,40,43]. Previous studies have found the expression of bone morphogenetic protein and transforming growth factor [20,21,22,23,24], along with increased fibronectin levels [21,22,23,24]. These findings suggest that OLF involves enchondral ossification. 

Although there is currently no consensus on the causes of cartilaginous ossification, previous studies have reported chronic mechanical stimulation to be a possible cause [7,8,21,24,25,27]. Therefore, from an anatomical and physiological perspective, the ligamentum flavum is present in the caudal half of the spinal canal at the vertebral arch surface. It is located at the upper edge of the lower vertebral joint capsule in the horizontal direction. It is composed of a large amount of elastic and collagen fibers. In the thoracic spine, the ligamentum flavum is always in an extended state, forming and maintaining a posterior bend. In the cervical and lumbar spines, alignment is anteriorly curved, and the ligamentum flavum pulls the vertebral arch with a tension that tends to contract the spine. The preferred sites are the transition points of curvature, the cervico-thoracic junction, and the lower thoracic spine, where both mechanical stimuli of extension and contraction apply [26,27]. In this study, increased elastic fiber proliferation was observed in the histopathological findings, suggesting that chronic mechanical stimulation (extension, traction, and rotation) and physiological function (contraction) of the ligamentum flavum can cause cartilaginous ossification in OLF.

Chronic stimulation (extension traction rotation and contraction) is believed to stimulate the epiphysis of the joints and induce enchondral ossification [7,8,21,24,25,27]. Recent studies on pathophysiology have shown that shearing stress to ligament fibers induces calcium deposition and the production of the bone morphogenetic protein and transforming growth factor [59,60]. Furthermore, the presence of bone marrow ossification suggested the involvement of genes upstream to bone formation genes in the pathogenesis of OLF.

### 4.5. Study Limitations

#### 4.5.1. Prevalence

In the current investigation, we were unable to determine the true prevalence of CLF and OLF. This study focused on patients in whom surgical intervention was performed due to neurological symptoms and signs arising from spondylotic changes in the spine at MIMH and OMUH between April 2010 and March 2022. Both CLF and OLF were evaluated based on specimens obtained from these cases. As a result, asymptomatic cases were not included. Furthermore, cases involving CLF and OLF with complications of cervical spondylosis, disc herniation, or lumbar spinal canal stenosis were excluded.

#### 4.5.2. Chronological Evidence

In the current study, we are unable to examine the temporal progression of changes. It is unclear whether calcification progresses to ossification, and we could not identify inducers of such potential progression. Some cases with OLF demonstrated local calcification independent of ossification. However, it remains unclear whether calcification and ossification coexist or whether a transitional phase occurs from calcification to ossification.

## 5. Conclusions

We clarified the differences between CLF and OLF. These differences should be considered when making therapeutic decisions, particularly related to surgical management. We should apply surgical procedures (laminectomy, en-block laminoplasty, or laminectomy with OLF), considering extending and adherent to the dura mater of calcification and ossification. 

During surgery for CLF, the ligamentum flavum, including CLF, should be removed en-block with the lamina. During surgery for OLF, special attention should be paid to the removal of OLF, particularly in cases of adhesions or penetration into the dura mater. Timely diagnosis and appropriate surgical intervention can yield favorable results in both CLF and OLF.

In the present and previous studies, we found that the pathogenesis of CLF involves inflammation and subsequent repair, primarily within ligamentous fibers, which induces metastatic and dystrophic calcification. Conversely, the pathogenesis of OLF involves enchondral ossification by the involvement of genes upstream to bone formation genes.

## Data Availability

All data are available on request to the corresponding author. The data are not publicly available due to privacy.

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
