# Peer review of "The Symptomatic Calcification and Ossification of the Ligamentum Flavum in the Spine: Our Experience and Review of the Literature"

_jcm, 2023, doi:10.3390/jcm13010105_

Round 1
Reviewer 1 Report
Comments and Suggestions for Authors
Overall well written
Extensively covered the topic
Pls describe the purpose of the current study in the intro and 1st para in discussion
Study design? Prospective or retrospective
Pls elaborate previously published literature comparing CLF and OLF
Pls present images illustrating the surgical intervention
Pls elaborate the limitations of the study
The conclusion may be more clearly (instead of going back and forth between CLF and OLF)
Comments on the Quality of English Language
Overall well written.
Minor corrections are needed
Author Response
Manuscript ID: jcm-2697896
Dear Editor and Reviewer 1
We deeply sincerely appreciate your review of our manuscript. The manuscript has been by a native English speaker.
We have revised the manuscript according to your suggestions as below. Please see my responses below.
We have the red highlight where we added and revised text.
Certification
The attached paper has been carefully reviewed by an experienced medical editor whose first language is English and who specializes in editing papers written by physicians and scientists whose native language is not English.
1: We have revised the Background, Introduction, and Discussion and the red highlighted the revised text (below).
Introduction
We retrospectively encountered 15 cases of CLF and 20 cases of OLF that required surgical treatment and histopathological analysis of excised specimens. In this study, we retrospectively reviewed our cases as well as the previous literature to compare CLF and OLF in terms of their clinical, radiological, and histopathological features and pathogenesis. In this study, we compared the distinct characteristics of CLF and OLF to enhance our understanding of these conditions, particularly especially in the context of therapeutic considerations, such as surgical management.
Discussion
4.2. Comparison of neuroradiological findings between CLF and OLF (Table 3; Figures. 3–6)
A previous study compared CLF and OLF, without considering MR imaging. Furthermore, changes in CT appearance over time were not evaluated, which prevented detailed analysis of the differences between calcification and ossification, as well as their anatomical relationships [12]. In the present study, we compared the neuroradiological features of CLF and OLF.
4.4. Pathogenesis of CLF and OLF based on histopathological findings (Table 5, Figures 7 and 8)
We compared the histopathologocal findings between CLF and OLF, including molecular biological characteristics. Conversely, the previous study did not examine the molecular biological characteristics [12]. In this study, we compared the pathological features of CLF and OLF.
2: I added in the 3rd. paragraph in the introduction that this study was the retrospective study.
3: I elaborated the previous published literature comparing CLF and OLF in the first paragraph in the Discussion (below).
4.2. Comparison of neuroradiological findings between CLF and OLF (Table 3; Figures. 3–6)
A previous study compared CLF and OLF, without considering MR imaging. Furthermore, changes in CT appearance over time were not evaluated, which prevented detailed analysis of the differences between calcification and ossification, as well as their anatomical relationships [12]. In the present study, we compared the neuroradiological features of CLF and OLF.
4: I added Figure 1 and 2 as the illustrates of the operations and legends.
Figure 1. 3D and 2D-CT images of CLF. 3D-reconstructed CT image showing changes after en-bloc expansive laminectomy and laminoplasty using basket type titanium plates (arrows) (A) and axial image (B). 3D-reconstructed CT image showing changes after single-staged posterior decompression (asterisks) and posterior lateral fixation using lateral mass screws and rods (arrows) (C) and axial image (D).
Figure 2. Illustrations demonstrating laminectomy for OLF in the thoracic spine. After thinning the lamina (black cross), cranial, caudal, lateral, and midline garters were made by drilling (black asterisks) (A and A’). The OLF was carefully dissected from the dura matter and removed (black arrows) (B and B’).
5: I added limitation of this study the last of discussion.
4.5. Study limitations
4.5.1. Prevalence
In the current investigation, we were unable to determine the true prevalence of CLF and OLF. This study focused on patients in whom surgical intervention was performed due to neurological symptoms and signs arising from spondylotic changes in the spine at MIMH and OMUH between April 2010 and March 2022. Both CLF and OLF were evaluated based on specimens obtained from these cases. As a result, asymptomatic cases were not included. Furthermore, cases involving CLF and OLF with complications of cervical spondylosis, disc herniation, or lumbar spinal canal stenosis were excluded.
4.5.2. Chronological evidence
In the current study, we are unable to examine the temporal progression of changes. It is unclear
whether calcification progresses to ossification, and we could not identify inducers of such potential progression. Somecases with OLF demonstrated local calcification independent of ossification. However, it remains unclear whether calcification and ossification coexist or a transitional phase occurs from calcification to ossification.
6: I totally revised and cleared the conclusion
5. Conclusions
We clarified the differences between CLF and OLF. These differences should be considered when making therapeutic decisions, particularly related to surgical management.
In the present and previous studies, we found that the pathogenesis of CLF involves inflammation and subsequent repair, primarily within ligamentous fibers, which induces metastatic and dystrophic calcification. Conversely, the pathogenesis of OLF involves enchondral ossification by the involvement of genes upstream to bone formation genes.
During surgery for CLF, the ligamentum flavum, including CLF, should be removed en-block with the lamina. During surgery for OLF, special attention should be paid to the removal of OLF, particularly in cases of adhesions or penetration into the dura mater. Timely diagnosis and appropriate surgical intervention can yield favorable results in both CLF and OLF.
Reviewer 2 Report
Comments and Suggestions for Authors
It was interesting to compare two diseases that require differentiation, which may seem similar at first glance, in many ways. Thank you for collecting good data and organizing it into results. However, there appear to be no new findings in this review paper compared to previous research results. Additionally, Errors in English words are frequently observed, and the level of completeness is disappointing.
page 2, Line 59, 60
Line break error, check comma period
page 5, line 149
spacing error
page 7, line 179
Caser --> Case
What means the words 'Case 1' and '6'?
Table 3.
Radiokogical --> Radiological
Page 7, Line 188, 189, 192
Iproved --> improved
Page 9, Figure 5, line 237 to 239
Pont size error
at C, D, please using the arrow or arterisk for identify the explanation
page 11, line 302, 303, 305, 313, 316
mtor --> motor
compltely --> completely
imprpvement --> improvement
Comments on the Quality of English Language
Errors in English words are frequently observed, and the level of completeness is disappointing.
Author Response
Manuscript ID: jcm-2697896
Dear Editor and Reviewer 2
We deeply sincerely appreciate your review of our manuscript. The manuscript has been by a native English speaker.
We have revised the manuscript according to your suggestions as below. Please see my responses below.
We have the red highlight where we added and revised text.
Certification
The attached paper has been carefully reviewed by an experienced medical editor whose first language is English and who specializes in editing papers written by physicians and scientists whose native language is not English.
For Reviewer 2:
1: We have revised the Introduction, Discussion, and Conclusion. I demonstrated the red highlighted in the revised text (below). Although this article has no new data, I review the previous data and report and validated the based on our own experiences.
1. Introduction
We retrospectively encountered 15 cases of CLF and 20 cases of OLF that required surgical treatment and histopathological analysis of excised specimens. In this study, we retrospectively reviewed and validated our cases as well as the previous literature to compare CLF and OLF in terms of their clinical, radiological, and histopathological features and pathogenesis. In this study, we compared the distinct characteristics of CLF and OLF to enhance our understanding of these conditions, particularly especially in the context of therapeutic considerations, such as surgical management.
- Discussion
4.4. Pathogenesis of CLF and OLF based on histopathological findings (Table 5, Figures 7 and 8)
We compared the histopathologocal findings between CLF and OLF, including molecular biological characteristics. Conversely, the previous study did not examine the molecular biological characteristics [12]. In this study, we compared the pathological features of CLF and OLF.
5. Conclusions
We clarified the differences between CLF and OLF. These differences should be considered when making therapeutic decisions, particularly related to surgical management.
In the present and previous studies, we found that the pathogenesis of CLF involves inflammation and subsequent repair, primarily within ligamentous fibers, which induces metastatic and dystrophic calcification. Conversely, the pathogenesis of OLF involves enchondral ossification by the involvement of genes upstream to bone formation genes.
During surgery for CLF, the ligamentum flavum, including CLF, should be removed en-block with the lamina. During surgery for OLF, special attention should be paid to the removal of OLF, particularly in cases of adhesions or penetration into the dura mater. Timely diagnosis and appropriate surgical intervention can yield favorable results in both CLF and OLF.
2: At Figure 5, we put the arrows and asterisks for identified the explanation.
Others
We had done spell checking.
Reviewer 3 Report
Comments and Suggestions for Authors
General comments:
The manuscript adresses the calcifications/ossifications of the PLL and lig. flavum in large series with imaging and histological correlation as well as long-term follow-up.
There are however several issues which have to be adressed to improve the validity of the m/s. They are stated in specific comments.
Also, there are many spelling mistakes.
Specific comments:
L. 76 Was the myelopathy/radiculopathy defined only clinically (Table 2) or also on the MRI? Based on what do authors state in the lines123 and 130 that »All cases developed myelopathy at the lesion site.«
L. 101, 270, 279. Exlain: »Of the 1,800 cases who had neurological symptoms and spinal surgery for spondylotic changes, 15 cases (0.8%) were cases.« Where does the No. 1.800 come from?
L. 112. Explain: »Of the 1,800 cases who had neurological symptoms and spinal surgery for spondylotic change, 22 cases (1.1%) had OLF.«
L. 125, 132, 192, 210 Correct »Brown sequred«
L. 131 Correct »Tetra paresis«
L. 138. How was Developmental narrowing of the spinal canal defined?
L. 148. Insert Figure 1 and Figure 2 in the main text.
L. 150. »posterior osteo-spur at the C5/6 intervertebral level« is not annotated with the arrow/arrowhead.
L. 154. Considering the definitions and A, A', B, B', on the image C there are diffuse and speck like calcification on the right side in C and along the PLL in C'. There egg shape calcifications on the left side on C and along the LF on C'.
L. 160. Figure 2. Explain »those mass lesions«. Is this the same patient as in the Figure 1? Also, correct »iamges«.
L, 173. correct »recionstructed«
L. 180. Table 3. At the MRI description of CLF »egg shape« and »speck like« changes are listed, whereas at the Figures only »low intensity« masses are described. In fact, I don't agree that one can distinguish between »speck like« and »egg shape« masses on MRI therefore »low intensity« is more correct description. I the authors can argument otherwise, please do so.
Comments on the Quality of English Language
There are many spelling mistakes, otherwise it is written in good English.
Author Response
Manuscript ID: jcm-2697896
Dear Editor and Reviewer 3
We deeply sincerely appreciate your review of our manuscript. The manuscript has been by a native English speaker.
We have revised the manuscript according to your suggestions as below. Please see my responses below.
We have the red highlight where we added and revised text.
Certification
The attached paper has been carefully reviewed by an experienced medical editor whose first language is English and who specializes in editing papers written by physicians and scientists whose native language is not English.
For Reviewer 3:
We have revised as the red highlighted tin he revised, table and figure according to your suggestions and advices.
1: Myelopathy/radiculopathy were defined by neurological signs. I added that in the manuscript (below).
2.2.2. Surgical indication and procedure
Surgery was indicated for patients with neurological signs of progressive myelopathy and/or radiculopathy.Posterior decompression by laminectomy or laminoplasty was performed under somatosensory evoked potential monitoring [46,47].
2: I added that in Results of the manuscript (below).
- Results
Of our 1,800 patients who presented to MIMH and OMUH between April 2010 and March 2022 for spinal surgery due to neurological symptoms and spondylotic changes, 15 (0.8%) had CLF.
Of our 1,800 patients who presented to MIMH and OMUH between April 2010 and March 2022 for spinal surgery due to neurological symptoms and spondylotic changes, 22 (1.2%) had OLF.
3: I changed.
4: I confirmed.
5: AP diameter 12mm>
3.3. Neuroradiological findings
CLF: All cases exhibited spondylotic changes. Developmental narrowing (anterior-posterior diameter of spinal canal < 12 mm)
4: I inserted Figure 1 and Figure 2 in the text.
5: I annotated the arrow heads in the Figure 3.
6: I revised the figure legend and the position of arrow heads in the Figure 3 C.
Figure 3. 2D-CT images of CLF. 2D reconstructed CT scan axial image (A) and sagittal image (A’) CT scan demonstrating “egg shape” high density mass contacting with laminae (arrows), posterior osteo-spur at the C5/6 intervertebral level (black arrows). 2D reconstructed CT scan axial image (B) and sagittal image (B’) demonstrating “diffuse and speck like” high density mass contacting with laminae in ligamentum flavum (white arrows). 2D reconstructed CT scan axial image (C) and sagittal image (C’) demonstrating “egg shape” high density mass in ligamentum flavum on the left side on C and along the ligamentum flavum on C’ (white arrows), ossification of posterior longitudinal ligament at the C4/5, 5/6 (white arrow heads) and listhesis at C3/4 intervertebral level. 2D reconstructed CT scan axial image (D) and sagittal image (D’) demonstrating “multiple islands of speck like and egg shape” high density mass in ligamentum flavum (arrows), and ossification of longitudinal ligament at C3/4 (white arrow head).
7: Those mass lesions are same patients Figure 1 and Figure 2., Figure 3 and Figure 4
8: I revised. Reconstructed
9: I revised Table 3.
Reviewer 4 Report
Comments and Suggestions for Authors
a. Authors state in the introdcution that CLF and OLF are distinct conditions as per some studies but doesn't provide specific details or key differences that set them apart.
b. The mention of "histopathogical" should be corrected to "histopathological."
c. In the statement that the pathogenesis of CLF is relatively less clear compared to other aspects of the condition might benefit from a brief explanation or example of what is known and what gaps remain
d. The introduction section implies that the study will review and compare CLF and OLF in various aspects but does not explicitly state the study's objectives or hypotheses
e. "magnetic resonace" should be "magnetic resonance."
f. "nenurological" should be "neurological."
g. The study period (April 2010 to March 2022) is quite long. It should be clarified whether there were any changes in treatment protocols or diagnostic criteria during this time.
h. For the surgical procedure, it's not clear if the same approach was used for all patients or if there were variations based on specific case requirements.
i. The discussion on OLF's relationship with DISH (Diffuse Idiopathic Skeletal Hyperostosis) and ASH (Ankylosing Spondylitis Hyperostosis) lacks specifics on how these associations were determined.
j. The section on the pathogenesis of CLF and OLF lacks clear differentiation between the two conditions in terms of their development and underlying mechanisms.
Author Response
Manuscript ID: jcm-2697896
Dear Editor and Reviewer 4
We deeply sincerely appreciate your review of our manuscript. The manuscript has been by a native English speaker.
We have revised the manuscript according to your suggestions as below. Please see my responses below.
We have the red highlight the newly added and revised text.
Certification
The attached paper has been carefully reviewed by an experienced medical editor whose first language is English and who specializes in editing papers written by physicians and scientists whose native language is not English.
For Reviewer 4:
a: We clarified specific details and key differences in the Discussion as the red highlighted the revised text (below).
CLF: CLF has been found to not reach the surface of the ligamentum flavum in the current and past reports [12–17]. Histopathologically, calcification resembling an island, speck, or grains of sand were observed. From a chronological perspective, these calcifications grew larger over time, coalescing into larger lesions. Previous reports showed the presence of HA in the center of the calcified lesion, whereas CPPD crystals were found in the surrounding area. These crystals exhibited cube, rhombus, or rod-shaped structures, ranging in size from 3 to 30 microns, and were partially engulfed by neutrophils and macrophages [5,6,8,9,11,15–17]. These findings suggest that the pathogenesis of CLF involves inflammation and subsequent repair process, primarily within ligamentous fibers. CPPD is involved in pseudogout and crystal-induced arthritis, suggesting that CLF is related to generalized calcinosis. Chondrocytes and osteoblasts were not observed in CLF, indicating that enchondral ossification does not occur.
Calcification is categorized into metastatic or dystrophic [43]. Metastatic calcification occurs in normal tissue, whereas dystrophic calcification occurs in necrotic tissue after inflammation [43]. Heterotopic (metastatic and dystrophic) calcification, such as CLF and flavum ossification of the longitudinal ligament, is considered to involve the conversion of progenitor cells to osteogenic precursor cells as a result of cell-mediated interactions with the local tissue environment as well as oxygen tension, pH, micronutrient availability, and heterotopic ossification due to mechanical stimuli [43]. The phenomenon of calcification requires one or more of least three phenomena without endochondral cell: (a) local increase in the concentration of precipitable ionic species (e.g., Ca,PO4); (b) the presence of “nucleator,” that is, a promotor of crystal deposition; or (c) the modification or removal of macromolecules inhibiting crystal deposition [43]. Previous studies have suggested that the pathogenesis of CLF involves aging-related changes, endocrine disorders, and chronic stimulation [43]. Mechanical stimulation, particularly traction and rotation, causes ligament fiber injury and subsequent calcification during the inflammatory repair process, primarily within ligamentous fibers. Inside the calcification, CPPD concentrates and HA is produced [7,8,11–17]. The lesion of calcification never connected to the epiphysis of the joints, and so the endochondral ossification never occurred. Because calcification progresses in focal reaction and cell unit in ligamentum flavum, calcification occurs surrounding area of the injured collagen fiber which is weak to tension and sharing stress and easy to be injured.
OLF: The radiological and histopathological findings demonstrate that laminar ossification occurs along with the elastic fibers within the ligament. Microhistopatholopgical findings showed adhesion and penetration into the dura matter, with ossification on the side of the dura mater[24,40,43]. Previous studies have found the expression of bone morphogenetic protein and transforming growth factor [20–24], along with increased fibronectin levels [21–24]. These findings suggest that OLF involves enchondral ossification.
Although there is currently no consensus on the causes of cartilaginous ossification, previous studies have reported chronic mechanical stimulation to be a possible cause [7,8,21,24,25,27]. Therefore, from an anatomical and physiological perspective, the ligamentum flavum is present in the caudal half of the spinal canal at the vertebral arch surface. It is located at the upper edge of the lower vertebral joint capsule in the horizontal direction. It is composed of a large amount of elastic and collagen fibers. In the thoracic spine, the ligamentum flavum is always in an extended state, forming and maintaining a posterior bend. In the cervical and lumbar spines, alignment is anteriorly curved, and the ligamentum flavum pulls the vertebral arch with a tension that tends to contract the spine. The preferred sites are the transition points of curvature, the cervico-thoracic junction, and the lower thoracic spine, where both mechanical stimuli of extension and contraction apply [26,27]. In this study, increased elastic fiber proliferation was observed in the histopathological findings, suggesting that chronic mechanical stimulation (extension, traction, and rotation) and physiological function (contraction) of the ligamentum flavum can cause cartilaginous ossification in OLF.
Chronic stimulation (extension traction rotation and contraction) is believed to stimulation the epiphysis of the joints and induce enchondral ossification [7,8,21,24,25,27]. Recent studies on pathophysiology have shown that shearing stress to ligament fibers induces calcium deposition and induces the production of the bone morphogenetic protein and transforming growth factor [58-61]. Furthermore, the presence of bone marrow ossification suggested the involvement of genes upstream to bone formation genes in the pathogenesis of OLF.
b: I changed and had done spell checking.
c: I revised the text about the pathogenesis of CLF (above) and limitation of this study about the pathogenesis in discussion as the red highlight (below).
4.5. Study limitations
4.5.1. Prevalence
In the current investigation, we were unable to determine the true prevalence of CLF and OLF. This study focused on patients in whom surgical intervention was performed due to neurological symptoms and signs arising from spondylotic changes in the spine at MIMH and OMUH between April 2010 and March 2022. Both CLF and OLF were evaluated based on specimens obtained from these cases. As a result, asymptomatic cases were not included. Furthermore, cases involving CLF and OLF with complications of cervical spondylosis, disc herniation, or lumbar spinal canal stenosis were excluded.
4.5.2. Chronological evidence
In the current study, we are unable to examine the temporal progression of changes. It is unclear
whether calcification progresses to ossification, and we could not identify inducers of such potential progression. Somecases with OLF demonstrated local calcification independent of ossification. However, it remains unclear whether calcification and ossification coexist or a transitional phase occurs from calcification to ossification.
d: I described the study’s objectives and hypothesis in the Introduction as the red highlight (below).
1. Introduction
Calcification of the ligamentum flavum (CLF) is a disease that spinal surgeon often encounter. Some studies have categorized CLF as distinct from the ossification of the ligamentum flavum (OLF) [1–11]. The clinical, radiological, and histopathological features and surgical management of CLF are well-known [1,2,4,7,11–16]. However, its pathogenesis is relatively less clear [4,5,7,8,12–15].
OLF is frequently encountered by spinal surgeons, and numerous previous studies have described its clinical and radiological features and surgical treatment [17–38]. Few studies had also described the histopathological features of OLF in the past [18,20-24,30], but the recent experimental studies have explored its pathogenesis using molecular and biological aspects [20–24,39–44].
We retrospectively encountered 15 cases of CLF and 20 cases of OLF that required surgical treatment and histopathological analysis of excised specimens. In this study, we retrospectively reviewed and validated our cases as well as the previous literature to compare CLF and OLF in terms of their clinical, radiological, and histopathological features and pathogenesis. In this study, we compared the distinct characteristics of CLF and OLF to enhance our understanding of these conditions, particularly especially in the context of therapeutic considerations, such as surgical management.
e: I changed.
f: I changed.
g: There was no change in the period of this study, and I added it as the red highlight in the methods (below).
2.2. Methods
2.2.1. Clinical features
Sex, age, prevalence, location, associated diseases were evaluated in each CLF and OLF. Pre- and post-operative neurological symptoms and signs were assessed. Neurological signs and activities of daily living (ADL) were evaluated using the Japanese orthopaedic association (JOA) score in cervical spondylotic myelopathy [45]. In thoracic and thoraco-lumbar spine lesion, the points for upper extremities were excluded, while in lumbar spine lesions, the points for upper extremities and trunk were excluded. This resulted in a maximum score of 17 points for cervical and cervico-thoracic spine lesions, 11 points for thoracic and thoraco-lumbar spine lesions and 9 points for lumbar spine lesions. The recovery rate (RR%) was calculated as follows: <(postoperative score − preoperative score)/(maximum score − preoperative score)> × 100 (%). Pre-operative whole spinal dynamic x-rays, two dimensional (2D) computed tomography (CT), magnetic resonance (MR) imaging were performed. Post-operatively neurological symptoms, signs and ADL were checked up, and spinal dynamic x-rays, 2D-CT, MR imaging of lesions were performed every 6 months. There was no change in the treatment protocol or diagnostic criteria between April 2010 and March 2022.
h: I added Figure 1 and 2, and its legends to clarify the surgical procedures.
i: I refer the reference and added the explanation as the red highlight in the discussion (below).
OLF: 2D-reconstruced CT scan images showed beak-like ossification extending into the intervertebral OLF: 2D-reconstruced CT scan images showed beak-like ossification extending into the intervertebral foramen. MR T1-weighted and T2-weighted images demonstrated a low-intensity mass within the dorsal spinal canal compressing the spinal cord. In some cases, the presence of high-intensity spots suggested bone marrow formation (Figures 3 and 4) [18,26,29,30,31]. In some cases, OLF was associated with DISH and ASH, indicating a more generalized ossification pattern (Table 3). Previous reports have reported the association of DISH and ASH in OLF, but the specific features of this association were unclear [51,52]. In both CLF and OLF, spondylotic changes were the most common associated condition, followed by disc herniation and instability. This suggested that chronic micro-movement and mechanical stress (traction, rotation, and compression) to the ligament were contributing factors (Table 3).
j: I cleared the differentiation between CLF and OLF in the Discussion as the red highlight (below).
4.4. Pathogenesis of CLF and OLF based on histopathological findings
We compared the histopathologocal findings between CLF and OLF, including molecular biological characteristics. Conversely, the previous study did not examine the molecular biological characteristics [12]. In this study, we compared the pathological features of CLF and OLF.
CLF: CLF has been found to not reach the surface of the ligamentum flavum in the current and past reports [12–17]. Histopathologically, calcification resembling an island, speck, or grains of sand were observed. From a chronological perspective, these calcifications grew larger over time, coalescing into larger lesions. Previous reports showed the presence of HA in the center of the calcified lesion, whereas CPPD crystals were found in the surrounding area. These crystals exhibited cube, rhombus, or rod-shaped structures, ranging in size from 3 to 30 microns, and were partially engulfed by neutrophils and macrophages [5,6,8,9,11,15–17]. These findings suggest that the pathogenesis of CLF involves inflammation and subsequent repair process, primarily within ligamentous fibers. CPPD is involved in pseudogout and crystal-induced arthritis, suggesting that CLF is related to generalized calcinosis. Chondrocytes and osteoblasts were not observed in CLF, indicating that enchondral ossification does not occur.
Calcification is categorized into metastatic or dystrophic [43]. Metastatic calcification occurs in normal tissue, whereas dystrophic calcification occurs in necrotic tissue after inflammation [43]. Heterotopic (metastatic and dystrophic) calcification, such as CLF and flavum ossification of the longitudinal ligament, is considered to involve the conversion of progenitor cells to osteogenic precursor cells as a result of cell-mediated interactions with the local tissue environment as well as oxygen tension, pH, micronutrient availability, and heterotopic ossification due to mechanical stimuli [43]. The phenomenon of calcification requires one or more of least three phenomena without endochondral cell: (a) local increase in the concentration of precipitable ionic species (e.g., Ca,PO4); (b) the presence of “nucleator,” that is, a promotor of crystal deposition; or (c) the modification or removal of macromolecules inhibiting crystal deposition [43]. Previous studies have suggested that the pathogenesis of CLF involves aging-related changes, endocrine disorders, and chronic stimulation [43]. Mechanical stimulation, particularly traction and rotation, causes ligament fiber injury and subsequent calcification during the inflammatory repair process, primarily within ligamentous fibers. Inside the calcification, CPPD concentrates and HA is produced [7,8,11–17]. The lesion of calcification never connected to the epiphysis of the joints, and so the endochondral ossification never occurred. Because calcification progresses in focal reaction and cell unit in ligamentum flavum, calcification occurs surrounding area of the injured collagen fiber which is weak to tension and sharing stress and easy to be injured.
OLF: The radiological and histopathological findings demonstrate that laminar ossification occurs along with the elastic fibers within the ligament. Microhistopatholopgical findings showed adhesion and penetration into the dura matter, with ossification on the side of the dura mater[24,40,43]. Previous studies have found the expression of bone morphogenetic protein and transforming growth factor [20–24], along with increased fibronectin levels [21–24]. These findings suggest that OLF involves enchondral ossification.
Although there is currently no consensus on the causes of cartilaginous ossification, previous studies have reported chronic mechanical stimulation to be a possible cause [7,8,21,24,25,27]. Therefore, from an anatomical and physiological perspective, the ligamentum flavum is present in the caudal half of the spinal canal at the vertebral arch surface. It is located at the upper edge of the lower vertebral joint capsule in the horizontal direction. It is composed of a large amount of elastic and collagen fibers. In the thoracic spine, the ligamentum flavum is always in an extended state, forming and maintaining a posterior bend. In the cervical and lumbar spines, alignment is anteriorly curved, and the ligamentum flavum pulls the vertebral arch with a tension that tends to contract the spine. The preferred sites are the transition points of curvature, the cervico-thoracic junction, and the lower thoracic spine, where both mechanical stimuli of extension and contraction apply [26,27]. In this study, increased elastic fiber proliferation was observed in the histopathological findings, suggesting that chronic mechanical stimulation (extension, traction, and rotation) and physiological function (contraction) of the ligamentum flavum can cause cartilaginous ossification in OLF.
Chronic stimulation (extension traction rotation and contraction) is believed to stimulation the epiphysis of the joints and induce enchondral ossification [7,8,21,24,25,27]. Recent studies on pathophysiology have shown that shearing stress to ligament fibers induces calcium deposition and induces the production of the bone morphogenetic protein and transforming growth factor [58-61]. Furthermore, the presence of bone marrow ossification suggested the involvement of genes upstream to bone formation genes in the pathogenesis of OLF.
Round 2
Reviewer 2 Report
Comments and Suggestions for Authors
authors made up for its shortcomings well.
Author Response
Manuscript ID: jcm-2697896
Dear Editor and Reviewer
We deeply sincerely appreciate your review of our manuscript. The manuscript has been by a native English speaker.
We have added the paragraphs in the revised manuscript according to the academic editor.
We have the red highlight where we added and revised text.
Certification
The attached paper has been carefully reviewed by an experienced medical editor whose first language is English and who specializes in editing papers written by physicians and scientists whose native language is not English.
In Methods: surgical indication and procedures
CLF: Conventional laminectomy or en-bloc laminectomy and laminoplasty using basket-type titanium plates were performed after dissection of the CLF (Figures 1A and 1B) [46–48]. For cases with instability, single-stage posterior decompression (laminectomy) and posterior lateral fixation (laminectomy with posterior lateral fixation [PLF]) using instrumentation was performed (Figures 1C and 1D) [49]. In most of the cases of CLF, there was no adhesion between the ligamentum flavum and the dura mater, and calcification did not extend to the intervertebral foramen. Therefor en-block laminoplsaty was generally considered the primary choice. For instances where calcification was extensive, as when calcification extended to the intervertebral foramen, laminectomy and posterior was performed.
OLF: In the cases of OLF, instances where the ligamentum flavum was adhered to the dura mater were not uncommon, and ossification extending to the intervertebral foramen was also frequently observed. Therefore, we had adopted laminectomy following the approach outlined in Figure 2. In cases who ossification did not extend to the intervertebral foramen nor adhered to the dura mater, we attempted en-block laminoplasty, considering the need to reconstruct posterior supporting tissue for long-term outcomes. A gutter with a similar width to the spinal canal was created, and the OLF was drilled out from the laminae. In cases where OLF was severely adherent to the dura mater, part of the OLF was intentionally left behind (Figure 2) [50,51]. For cases with instability, single-stage laminectomy with PLF was performed [49].
In Discussion: surgical treatments and outcome
CLF: In most of CLF cases, to prevent post-operative kyphosis in cases who calcification did not extend to the intervertebral foramen nor adhered to the dura mater, we attempted en-block laminoplasty. The ligamentum flavum, including CLF, can be removed without damage through en-block laminoplasty because the calcification is present inside the ligament and does not reach the surface of the ligament or cause adherence to the dura mater. However, caution is required when excising the ossified portion of the ligamentum flavum from the dura mater, as illustrated in Figure 2. In some instances, we deliberately left the calcification thin and chose not to detach it from the dura mater.
OLF: In OLF, as operative procedure, laminectomy was generally indicated. But we think that to prevent postoperative kyphosis we should try to perform laminoplasty for the possible case who ossification dose not extent to the intervertebral foramen nor adhered to the dura mater as illustrated in Figure 5B and 5B’). Since most of cases of OLF exhibit adhesion to and penetration into the dura mater, caution is required during OLF removal as illustrated in Figure 2. In cases with adhesion to the dura mater, it may be necessary to carefully preserve the OLF or suture defects in the dura mater after the dissection of the OLF (Figure. 2) [49,51].
In Conclusion
We clarified the differences between CLF and OLF. These differences should be considered when making therapeutic decisions, particularly related to surgical management. We should applicate surgical procedures (laminectomy, en-block laminoplasty or laminectomy with OLF, considering extending and adherent to the dura mater of calcification and ossification.
Reviewer 3 Report
Comments and Suggestions for Authors
The comments of the first review were reasonably included in the revised version.
Author Response

(The authors gave the same response as above.)

Reviewer 4 Report
Comments and Suggestions for Authors
Authors have answered all the concerns.
Author Response

(The authors gave the same response as above.)
